# Antagonistic Efficacy of Luteolin against Lead Acetate Exposure-Associated with Hepatotoxicity is Mediated via Antioxidant, Anti-Inflammatory, and Anti-Apoptotic Activities

**DOI:** 10.3390/antiox9010010

**Published:** 2019-12-21

**Authors:** Wafa A. AL-Megrin, Afrah F. Alkhuriji, Al Omar S. Yousef, Dina M. Metwally, Ola A. Habotta, Rami B. Kassab, Ahmed E. Abdel Moneim, Manal F. El-Khadragy

**Affiliations:** 1Biology Department, Faculty of Science, Princess Nourah bint Abdulrahman University, Riyadh 11671, Saudi Arabia; wafa.megren@gmail.com; 2Department of Zoology, Faculty of Science, King Saud University, Riyadh 11451, Saudi Arabia; aalkhuriji@ksu.edu.sa (A.F.A.); syalomar@KSU.EDU.SA (A.O.S.Y.); mdbody7@yahoo.com (D.M.M.); 3Department of Parasitology, Faculty of Veterinary Medicine, Zagazig University, Zagazig 44519, Egypt; 4Department of Forensic Medicine and Toxicology, Faculty of Veterinary Medicine, Mansoura University, Mansoura 35516, Egypt; ola_ali@mans.edu.eg; 5Department of Zoology and Entomology, Faculty of Science, Helwan University, Cairo 11795, Egypt; rami.kassap@yahoo.com (R.B.K.); aest1977@hotmail.com (A.E.A.M.)

**Keywords:** oxidative stress, apoptosis, inflammation, lead acetate, luteolin, liver

## Abstract

The abundant use of lead (Pb; toxic heavy metal) worldwide has increased occupational and ecosystem exposure, with subsequent negative health effects. The flavonoid luteolin (LUT) found in many natural foodstuffs possesses antioxidant and anti-inflammatory properties. Herein, we hypothesized that LUT could mitigate liver damage induced by exposure to lead acetate (PbAc). Male Wistar rats were allocated to four groups: control group received normal saline, LUT-treated group (50 mg/kg, oral, daily), PbAc-treated group (20 mg/kg, i.p., daily), and LUT+PbAc-treated group (received the aforementioned doses via the respective routes of administration); the rats were treated for 7 days. The results revealed that PbAc exposure significantly increased hepatic Pb residue and serum activities of aspartate aminotransferase (AST), alanine aminotransferase (ALT), and total bilirubin value. Oxidative reactions were observed in the liver tissue following PbAc intoxication, characterized by the depletion and downregulation of antioxidant proteins (glutathione, glutathione reductase, glutathione peroxidase, superoxide dismutase, catalase, nuclear factor erythroid 2-related factor 2, and heme oxygenase-1), and an increase in oxidants (malondialdehyde and nitric oxide). Additionally, PbAc increased the release and expression of the pro-inflammatory cytokines (tumor necrosis factor alpha and interleukin-1 beta), inducible nitric oxide synthase, and nuclear factor kappa B. Moreover, PbAc enhanced hepatocyte loss by increasing the expression of pro-apoptotic proteins (Bax and caspase-3) and downregulating the anti-apoptotic protein (Bcl-2). The changes in the aforementioned parameters were further confirmed by noticeable histopathological lesions. LUT supplementation significantly reversed all of the tested parameters in comparison with the PbAc-exposed group. In conclusion, our findings describe the potential mechanisms involved in the alleviation of PbAc-induced liver injury by luteolin via its potent anti-inflammatory, antioxidant, and anti-apoptotic properties.

## 1. Introduction

The liver is a vital organ involved in many significant biological functions, such as nutritional balance, glucose and cholesterol metabolism, and synthesis of clotting factors. Owing to its substantial blood supply, the liver is responsible for metabolizing potentially toxic xenobiotics, rendering it vulnerable to toxic injury [1]. Lead (Pb) is a non-essential persistent metal with colorless, tasteless, and odorless characteristics. It persists in the environment for a long time and can be detected at harmful concentrations in the environment and living organisms [2]. Exposure to Pb occurs through contaminated drinking water, battery production, lead paints, lead containing gasoline, and industrial emissions [3]. After absorption, Pb undergoes hepatic conjugation and is delivered to the kidneys, through which small amounts are excreted via urine; however, the majority of Pb accumulates in various organs. Post-mortem examination of human bodies has revealed that the liver is the main storehouse of Pb, storing approximately one third of the total absorbed lead, followed by the kidney [4].

Lead is a multi-organ toxicant that affects vital organs, such as the brain, liver, kidney, and testis, causing various diseases and cancers [5,6,7,8,9,10]. Hepatotoxicity induced by Pb was previously reported to be related to the destruction of liver structure and loss of function [11,12]. Chelating agents that are currently used for the treatment of Pb intoxication cannot remove the intracellular metal. Furthermore, they cannot reverse the associated side effects, such as toxicant redistribution, loss of essential metals, and liver or renal malfunction [13]. However, in recent times, use of the active constituents of some plants for treating toxicity has increased considerably. Flavonoids are polyphenolic compounds naturally present in the form of glycosides in many fruits and vegetables and have a variety of beneficial health effects [14,15]. Luteolin (3,4,5,7-tetrahydroxyflavone, LUT) is a flavonoid found in parsley, carrots, onion leaves, green peppers, celery, apple skins, honeysuckle blooms, and chamomile blossoms [16]. The remarkable effects of luteolin in the alleviation of inflammation, allergies, cancers, anxiety, heart diseases, and oxidative insults have been previously reported [17,18,19,20,21]. Luteolin exerts defensive effects against liver injury caused by acetaminophen, mercuric chloride, and tetrachloromethane by restoring antioxidant power and mitigating the inflammatory mediators [22,23,24]. Furthermore, LUT has shown significant anti-peroxidative effects against CCl_4_-induced hepatotoxicity in rats [25]. Therefore, the current study addresses the potential protective effects of pre-treatment with LUT on the oxidative events, inflammation, and apoptotic signaling pathways in liver tissue induced by Pb-intoxication in male Wistar rats.

## 2. Material and Methods

### 2.1. Chemicals and Reagents

Lead (II) acetate trihydrate (Pb(CH_3_CO_2_)_2_ 3H_2_O; CAS Number 6080-56-4, was obtained from Sigma-Aldrich (St. Louis, MO, USA). Luteolin (C_19_H_18_O_6_; CAS number 855-97-0) was purchased from Alfa Aesar (Kandel, Germany). Luteolin was first dissolved in dimethyl sulfoxide (DMSO) and then diluted with normal saline (0.9% NaCl); the final administered solution contained 5% DMSO, corresponding to 20 μL per rat. Control rats received the same dose of DMSO (5% DMSO in saline). Although DMSO could represent free radical scavenger effect, its use was necessary because Luteolin is hydrophobic. It was therefore used at the lowest level possible. The lead acetate was dissolved in double-distilled water. All other chemicals and reagents used were of analytical grade.

### 2.2. Animals and Experimental Ethics Protocol

Twenty-eight male Wistar albino rats, 10 weeks old and weighing 180–220 g, were used for the experimental procedures. They were obtained from VACSERA (Giza, Egypt). Prior to the experiment, the rats were housed in wire polypropylene cages under controlled environmental conditions (25 ± 2 °C temperature, 55–60% humidity and a 12 h light/dark cycle) for 2 weeks. All rats were fed a standard diet with free access to water. The experimental protocol was conducted in accordance with the European Community Directive (86/609/EEC). The animal care procedures agreed with the National Institutes of Health (NIH) Guidelines for the Care and Use of Laboratory Animals, 8th edition, and were approved by the Institutional Animal Ethics Committee for Laboratory Animal Care at the Zoology Department, Faculty of Science, Helwan University (Approval Number: HU/Z/012-19).

### 2.3. Experimental Design and Sampling

After the acclimation period, rats were randomly assigned to one of four groups (*n* = 7 per group). Group I (control) received orally administered 3 mL/kg per day of normal saline solution (0.9% NaCl), followed by an intraperitoneal injection (i.p.) of 1 mL/kg per day of saline after 3 h. Group II (LUT) received orally administered LUT at a dose of 50 mg/kg per day, daily at 10:00 AM, according to the method described by Kalbolandi et al. [26]; then, rats were i.p. injected with 1 mL/kg per day of saline after 3 h. Group III (lead acetate; PbAc) received orally administered 3 mL/kg per day of normal saline solution; then, PbAc at a dose of 20 mg/kg per day was injected i.p., as described by Abdel Moneim [10]. In Group IV (LUT+PbAc), LUT was orally administrated 3 h before the i.p. injection of PbAc. The treated animals were observed daily during the experimental period of seven days for any signs of toxicity.

Twenty-four hours after the last treatment, all animals were sacrificed by cervical dislocation after sodium pentobarbital (Sigma-Aldrich) at a dose of 300 mg/kg. Blood was obtained from the abdominal aorta using a syringe and the serum was separated. The liver tissue was carefully sampled, weighted, and washed twice in ice-cold 50 mM Tris–HCl of neutral pH. Immediately after weighing, the liver tissue was divided into three parts. The first was mixed with ice-cold medium containing 50 mM Tris–HCl (pH 7.4) and separately centrifuged at 3000× *g* for 10 min at 4 °C to prepare the tissue homogenate (10 % w/v). Then, the obtained supernatants were utilized for biochemical analyses. The second part was stored at −80 °C until use in the quantitative real time PCR test. The third part was utilized for the histopathological screening of hepatic tissue. Other organs were collected including brain, kidney, and testis for further studies.

### 2.4. Hepatic Lead Concentration

In accordance with the method described by Szkoda and Zmudzki [27], liver samples (2–10 g) were oven-dried at 120 °C overnight, and then, placed in a cool muffle furnace with the temperature gradually increased to 450 (50 °C/h). After cooling, concentrated nitric acid was used for sample digestion. Subsequently, the samples were returned to the muffle furnace at 450 °C/h, and then, allowed to cool. The dried ash was dissolved in 1 N HCl (1 g ash in 1 mL of HCl). The final solutions were diluted in 0.2% nitric acid. The actual hepatic lead concentrations, as µg/g wet tissue weight, were determined at 283.3 nm by flame atomic absorption spectrophotometry (Perkin-Elmer, 3100; Perkin-Elmer Corporation, Norwalk, CT, USA).

### 2.5. Liver Function Parameters

Liver functions tests on serum samples were calorimetrically conducted using standard kits following the manufacturer’s instructions. The enzymatic activities of aspartate aminotransferase (AST) and alanine aminotransferase (ALT), as well as the total bilirubin levels were estimated using commercial kits sourced from Biodiagnostic, Giza, Egypt.

### 2.6. Hepatic Inflammatory Biomarkers

The concentrations of tumor necrosis factor (TNF)-α and interleukin (IL)-1β in the liver homogenates were assessed using ELISA kits (R&D System, Minneapolis, MN, USA) following the manufacturers’ instructions. 

### 2.7. Hepatic Apoptotic Biomarkers

The protein levels of Bax, caspase 3, and Bcl-2 in the liver homogenates were determined using ELISA kit provided by Cusabio (Wuhan, China) according to the manufacturer’s manual protocols. 

### 2.8. Hepatic Oxidative Stress Markers

Lipid peroxidation (LPO) in liver tissues was assayed spectrophotometrically in liver homogenates by measuring malondialdehyde (MDA), a secondary product of lipid peroxidation, according to the procedure described by Ohkawa et al. [28]. Nitrite/nitrate (nitric oxide; NO) was estimated calorimetrically using the Griess reagent following the method of Green et al. [29]. Glutathione (GSH) was assayed spectrophotometrically at 405 nm according to the method of Ellman [30] based on the reduction of 5,5′-dithiobis (2-nitrobenzoic acid) to yellow-colored 5-thionitrobenzoic acid by glutathione.

### 2.9. Liver Antioxidant Capacity

Superoxide dismutase (SOD) activity was evaluated depending on its capacity to inhibit the reduction of the nitroblue tetrazolium dye to diformazan, which is mediated by phenazine methosulphate (PMS), according to the description by Sun et al. [31]. Catalase (CAT) activity was evaluated on the basis of the conversion of hydrogen peroxide (H_2_O_2_) to water and oxygen at 240 nm as shown by Aebi [32]. Assessment of glutathione peroxidase (GPx) and glutathione reductase (GR) activities was conducted following the methods of Paglia and Valentine [33] and Carlberg and Mannervik [34], respectively.

### 2.10. Gene Expression Analysis

The molecular mechanisms of oxidative injury, inflammatory responses, and apoptosis induced by PbAc exposure were assessed using quantitative real-time polymerase chain reaction (qRT-PCR). Total RNA was isolated using the TRIzol reagent (Life Technologies, Gaithersburg, MD, USA) according to the manufacturer’s guidelines. Then, cDNA was immediately synthetized using the MultiScribe RT enzyme kit (Applied Biosystems, Foster City, CA, USA). The obtained cDNA was subjected, to real-time PCR analysis. Real-time PCR reactions were accomplished using Power SYBR Green PCR Master Mix (Applied Biosystems, Life Technologies, Foster, CA, USA) on a 7500 Real-Time PCR System (Applied Biosystems, Foster City, CA, USA). The thermal cycle for the PCR analysis was 95 °C for 4 min, 40 cycles at 95 °C for 10 s, 60 °C for 30 s, and 72 °C for 10 s. The relative fold changes in mRNA expression of *Sod2*, *Cat*, *Gpx1*, *Gsr*, *Nfe2l2, Hmox1, Il1b*, *Tnfa*, *Nos2*, *Casp3*, *Bax*, and *Bcl_2_* genes were determined and compared to those of the control. Glyceraldehyde-3-phosphate dehydrogenase (*Gapdh*) was utilized as a reference housekeeping gene. Primer sequences and accession numbers of the genes are provided in Table 1. Experiments were performed as five assays in duplicate.

### 2.11. Immunohistochemical Examination

For immunohistochemistry examination of nuclear factor kappa B (NF-κB) in the hepatic tissue, a standard protocol with monoclonal anti-NF-κB antibody (Santa Cruz, CA, USA) was employed. Hepatic sections were examined using a 400× magnification lens (Nikon Eclipse E200-LED, Tokyo, Japan).

### 2.12. Western Blotting Analyses

Protein extraction and blotting analyses were carried out as previously reported [35]. The utilized antibodies included mouse antibody to Nrf2 (MAB3925, 1:500; R&D System), HO-1 (sc-390991, 1:750; Santa Cruz Biotechnology, Santa Cruz, CA, USA), β-actin (MAB8929, 1:500; R&D System), and goat anti-mouse IgG (sc-2039, 1:5000; Santa Cruz Biotechnology, Santa Cruz, CA, USA). The proteins were visualized using an enhanced chemiluminescence detection kit (Bio-Rad, Hercules, CA, USA) following the manufacturer’s protocol. Images were analyzed using the Kodak Image Station 2000R (Eastman Kodak Company, Rochester, NY, USA). Protein bands intensity were referenced to β-actin, and the data presented in terms of percent relative to controls.

### 2.13. Histopathological Examination

Liver samples were fixed in 10% neutral buffered formalin for 24 h, dehydrated in ascending grades of ethyl alcohol, cleared with xylene, and embedded in molten paraplast. The resulting blocks were sectioned into 4–5-μm thick sections and stained with hematoxylin and eosin. The microscopic assessment of tissue sections was done using a Nikon microscope (Eclipse E200-LED, Tokyo, Japan).

### 2.14. Statistical Analysis

All values are expressed as the mean ± standard deviation (SD). Collected data were subjected to one-way analysis of variance (ANOVA) followed post hoc by Duncan’s multiple range tests to determine significant differences between groups. The statistical differences between groups were considered significant at *p*-values < 0.05.

## 3. Results

### 3.1. Clinical Signs and Mortality

The experimental rats treated with either PbAc or LUT did not show any abnormal clinical signs or suffer any mortality during the exposure period. Oral administration of LUT to rats did not affect their food and water consumption, whereas the PbAc-injected group exhibited a significant decline in food intake, resulting in a marked decrease in the animals’ body weight (Appendix A).

### 3.2. Lead Concentration in Liver Tissue

The Pb concentrations in hepatic specimens from the PbAc-injected and LUT+PbAc-treated groups showed distinct elevations (*p* < 0.05) relative to that from the control rats. Furthermore, pre-treatment of rats with LUT markedly lowered (*p* < 0.01) the Pb residue in the liver tissue, compared to that after sole treatment with PbAc (Figure 1).

### 3.3. Liver Function Markers

The indicators of liver integrity, including aspartate aminotransferase (AST) and alanine aminotransferase (ALT), and total bilirubin levels were measured in the liver samples. The injection of rats with PbAc lead to a significant rise (*p* < 0.05) in AST, ALT, and total bilirubin levels, compared to those of the control group, whereas prior treatment with LUT markedly lowered (*p* < 0.05) the levels of these physiological indices relative to those of the PbAc-treated group (Figure 2).

### 3.4. Antioxidant/Oxidant Capacity

In our study, 1 week after PbAc and/or LUT treatments, the protein levels and mRNA expression of the antioxidant molecules (SOD, CAT, GPx, and GR) and oxidants, including MDA and NO along with GSH, were investigated in the liver tissues. Significant decreases in the levels of SOD, CAT, GPx, and GR were recorded (*p* < 0.05), associated with downregulation of their gene expression, i.e., *Sod2*, *Cat*, *Gpx*, and *Gsr* in PbAc-treated rats, compared to the control values. These alterations were remarkably protected when LUT was administrated before PbAc, as LUT significantly boosted (*p* < 0.05) the antioxidant status of hepatic tissue and significantly increased (*p* < 0.05) GSH concentration. Treatment with PbAc produced a dramatic elevation in MDA and NO levels, along with a significant decrease in GSH, compared with those of the control group. These effects were markedly reversed (*p* < 0.05) when LUT was administered before the PbAc injection (Figure 3 and Figure 4).

To understand the molecular antioxidant mechanism of LUT against PbAc-induced oxidative damage in the liver tissue, the expression of Nrf2 and HO-1, which control and regulate the activity of antioxidant molecules, was examined. qRT-PCR findings showed significant downregulation of *Nfe2l2* and *Hmox1* expression upon PbAc intoxication, compared to that in the control group. Meanwhile, LUT pre-treated rats exhibited a significant upregulation of these antioxidant promoters, compared to that in PbAc-exposed rats (Figure 5). In consistence with qRT-PCR findings, blotting analyses showed a significant downregulation in Nrf2 and HO-1 expression following PbAc exposure and this effect was abolished in LUT+PbAc treated group as compared to PbAc exposed group (Figure 5).

### 3.5. Inflammatory Markers

To further study the inflammatory conditions in the liver tissue in response to PbAc and/or LUT, the protein levels and mRNA expression of pro-inflammatory cytokines (IL-1β and TNF-α), mRNA expression of *Nos2* and immunohistochemical examination of NF-κB were investigated. PbAc-intoxicated rats exhibited a significant increase (*p* < 0.05) in the levels of the tested inflammatory markers and upregulation of their gene expression, i.e., *Il1b* and *Tnfa* along with *Nos2*, in comparison with that of the controls. However, their levels were markedly diminished (*p* < 0.05) following pre-treatment with LUT, compared to that of the PbAc-treated rats, as shown in Figure 6. In addition, immunohistochemical examination showed that PbAc exposure increased the expression of NF-κB as compared to its expression in the control group, while the pretreatment with LUT decreased the expression of NF-κB in the liver tissue as compared to PbAc treated group (Figure 7).

### 3.6. Hepatic Apoptotic Parameters

The PbAc treatment significantly enhanced the protein levels and expression of pro-apoptotic parameters; caspase-3 (*p* < 0.05) and Bax (*p* < 0.05) were enhanced in the liver tissue, in comparison with the control. In contrast, the level and mRNA expression of Bcl-2, an anti-apoptotic marker, exhibited a significant decline (*p* < 0.05) following PbAc exposure. Pre-administration of LUT significantly suppressed the loss of hepatocytes by downregulating the pro-apoptotic proteins and enhancing the upregulation of the anti-apoptotic proteins, compared to that with PbAc-treatment alone (Figure 8).

### 3.7. Alterations in Hepatic Tissue Architecture

Normal hepatic histological architecture was observed in the control and LUT groups, featuring normal central veins surrounded by normal intact hepatocytes (Figure 9A,B). In contrast, PbAc-intoxicated rats exhibited necrotic liver cells associated with considerably degenerated and vacuolated peripheral hepatocytes, along with neutrophil and lymphocyte infiltration around the peri-portal areas (Figure 9C). However, the liver sections of rats pre-treated with LUT exhibited obvious protection, as indicated by no vacuolar degeneration of peripheral hepatocytes (intact hepatocytes) and mild cellular infiltration, with only a few lymphocytes around the portal areas (Figure 9D).

## 4. Discussion

In the current study, a significant rise in Pb residue was observed in the hepatic tissue of the Pb-exposed group. This increase was in accordance with previous results [36,37,38]. Pb accumulated primarily in the liver and kidneys for two reasons; firstly, the liver and kidneys are the main sites for the excretion of Pb and, secondly, they contain proteins, such as thymosin β4, metallothionein, and acyl-CoA binding protein, that have high affinity for Pb-binding [39]. Furthermore, the group exposed to dual treatment with LUT+PbAc exhibited a marked decline in hepatic Pb concentrations. A previous report speculated that luteolin may provide protection against Pb-induced health deficits through its promising chelating properties [40]. Flavonoids possess a metal-chelating ability and can bind with metals to form a metal-flavonoid complex. Their ability to metal-complexes formation is owing to the presence of functional groups, carbonyl and hydroxyl groups, which are attached to ring structures of flavonols [41]. Surprisingly, a previous in vitro study revealed that these metal-flavonoid chelates have significant water solubility, which plays a vital role in the acceleration of the detoxification process [42]. Abdel-Moneim et al. [36] found that curcumin, a plant constituent rich in flavonoids, significantly reduced renal Pb content. Additionally, co-treatment with the active constituent, luteolin, from *Vaccinium corymbosum* L. blueberries significantly reduced the Cd burden in murine hepatic tissue [43].

In support of previous reports, significant elevations in the levels of liver damage markers, including ALT, AST, and total bilirubin, following PbAc exposure were observed [38,44]. The increase in these markers resulted from changes in the membrane permeability of hepatocytes, which allowed these enzymes to leak into circulation [39,44]. The disturbance in hepatic membrane permeability could be attributed to the action of free radicals induced by PbAc, which increase lipid peroxidation levels, in addition to the competition of Pb with calcium, an essential element for cell membrane integrity [39]. Biochemical data confirm the histological damage of the liver tissue. Previous studies reported Pb-induced anemia caused by the inhibitory effect of Pb on δ-aminolevulinic acid dehydratase (ALAD), which plays a significant role in heme synthesis, in addition to the formation of basophilic stippling in red blood cells with short life spans [12,36]. The elevated levels of total bilirubin in the PbAc-treated group may be attributed to massive heme destruction and obstruction of the biliary tract, with subsequent suppression of the conjugation reaction and release of unconjugated or indirect bilirubin from damaged liver cells [45]. Pre-treatment with LUT significantly improved liver function biomarker levels and indicated that LUT could preserve hepatocyte integrity and relieve liver damage. This is consistent with the findings of Zhang et al. [46] and Yang et al. [22]. Tai et al. [23] reported that LUT treatment improved the disturbance in levels of liver function markers evoked by acetaminophen and that was evident by regeneration of liver cells and healing of damaged hepatic parenchyma. In our study, the histopathological screening of liver tissue showed that hepatic necrosis and degeneration convinced by PbAc exposure were attenuated by LUT pre-administration.

In accordance with previous data, exposure of rats to PbAc suppressed activities and downregulated mRNA expression of the antioxidant enzymes, i.e., GR, GPx, SOD, and CAT, in hepatic tissues [12,38,44,47,48]. Enhancement of reactive oxygen species (ROS) generation is considered to be the primary mechanism of the induction of hepatic stress by Pb. ROS has the ability to enhance lipid peroxidation and exhaust antioxidant power in the cell. Additionally, Pb exhibits high binding affinity with the sulfhydryl groups of many enzymatic and non-enzymatic antioxidants, which subsequently diminishes their actions [47]. Pb can compete with and substitute some essential ions, such as zinc and copper, that represent vital cofactors in SOD and CAT activities [49]. SOD exerts its antioxidant actions by converting the toxic superoxide ion into a less toxic H_2_O_2_ compound, whereas the CAT enzyme dissociates it into water and oxygen [12]. Glutathione (GSH) is the most important antioxidant in cells and represents the major defense tool against metal intoxication via its chelating capacity. Glutathione exists in both reduced (GSH) and oxidized (GSSG) forms. GSH can be restored from GSSG by the aid of the glutathione reductase enzyme [49]. GPx is located in cellular membranes and aids in the depletion of H_2_O_2_ by converting GSH into GSSG. Hence, the diminished hepatic GPx after PbAc injection may refer to its role in lipid peroxide dismutation or the interaction of Pb with some metals important for GPx activity, such as selenium [38]. The obtained results of high hepatic NO levels suggested that Pb can induce oxidative stress by the induction of reactive nitrogen species (RNS) formation; this is in agreement with the results of other studies [47,50]. Excess NO reacts with the superoxide anion (O_2_) and forms the peroxynitrite (ONOO^−^) radical, which is a potent biological oxidant and much more reactive than its parent compounds, NO and O_2_^−^• [51].

Nuclear factor erythroid 2-related factor 2 (Nrf2) is an important cytoprotective agent that regulates the production of endogenous anti-oxidative enzymes. Normally, Nrf2 is sequestered in the cytoplasm by Kelch-like ECH-associated protein-1 (Keap1). Under the unpleasant circumstances of oxidative stress, Nrf2 is liberated from Keap1, moves to the nucleus, and stimulates the antioxidant response element (ARE); in turn, there is an increase in the production of anti-oxidative enzymes, such as catalase, nitrite oxidase, and heme oxygenase-1 (HO-1) [52]. Moneim [47] found that exposure to PbAc downregulated the mRNA gene expression of Nrf2 in liver cells in male rats. This effect could be due to the over activation of nuclear factor kappa B, which elevates the level of Keap1, resulting in Nrf2 dysregulation and suppression of its related detoxifying and antioxidant molecules [53]. However, LUT administration prior to PbAc upregulated mRNA gene expression of Nrf2, indicating its antioxidant action, which is consistent with results previously reported by Yang et al. [22]. HO-1 catabolizes free heme into Fe^2+^, carbon monoxide, and biliverdin. During oxidative stress, free heme encourages the over-generation of free radicals. Cells stimulate HO-1, which enhances catabolism of free heme, to avoid its pro-oxidant impacts [54]. Similar to other reports, Pb exposure in the current study, inhibited mRNA expression of this enzyme, whereas dual treatment with LUT and Pb upregulated its expression; thus, this can be considered an additive mechanism of LUT’s cytoprotection against oxidative stress [55]. The current findings showed that Pb exposure resulted in overproduction of ROS. ROS at significant concentrations might attack the vital cellular components, such as lipids in the cell membrane, proteins, and nucleic acids. Lipid peroxidation refers to oxidative damage of cellular lipids in which the free radicals steal electrons from unsaturated fatty acids resulting in serious cellular and functional damage. Hepatocytes are very vulnerable to LPO due to the presence of phospholipids rich in polyunsaturated fatty acids in the cell membranes [39]. Rats injected with Pb exhibited high MDA content in liver homogenates, and these results were in agreement with the elevated ALT and AST activities and the findings reported by other studies [39,47,56].

Pre-treatment with LUT improved the Pb-induced oxidative damage by elevating SOD, CAT, GR, GPx, and GSH and lowering the MDA level in hepatic tissue. The antioxidant potential of LUT was previously illustrated by other authors [22,46,57]. Flavonoids, including LUT, exert their antioxidant properties through several mechanisms involving the ability to trap ROS via chain breaking properties, thereby exerting their defensive effects on cellular macromolecules, in addition to increasing the cellular level of total GSH and suppressing the peroxidation of lipids [57,58]. Additionally, LUT employs its antioxidant ability through scavenging both ROS and RNS, chelating transition metals such as iron and copper that may cause oxidative damage through the Fenton reaction, suppression of pro-oxidant enzymes and enhancing activities of antioxidant enzymes [59,60,61]. Furthermore, antioxidant ability of LUT has not only reported in vivo but also observed by previous authors in vitro [62,63].

In the current study, the potential inflammatory reaction induced by PbAc treatment was proven by the high levels of pro-inflammatory cytokines, IL-1β and TNF-α. These results are in agreement with earlier outcomes of Pb toxicity studies [12,37]. The molecular findings revealed upregulation of the mRNA expression of IL-1β and TNF-α along with overexpression of NF-κB in the hepatic tissue. Pb has the ability to activate the mitogen-activated protein kinase (MAPK) pathway, which plays a vital role in the generation of pro-inflammatory cytokines, such as TNF-α. Additionally, Pb is involved in activation of NF-κB, which controls the expression of many genes involved in inflammation, such as pro-inflammatory cytokines, chemokines, and adhesion molecules [12,64]. Earlier investigations found that LUT administration relieved inflammatory reactions by decreasing the levels of pro-inflammatory cytokines and their mRNA expression in liver tissue [17,18]. Previously, LUT was reported to antagonize the mercury-induced inflammatory response via suppression of the MAPK pathway and reduction in the activity of NF-κB [22]. Other anti-inflammatory mechanisms involve the suppression NF-κB by LUT. Additionally, LUT can inhibit the activities pro-inflammatory enzymes, such as cyclooxygenases (COX), lipoxygenases (LOX), and inducible nitric oxide synthase (iNOS), that participate in the synthesis of eicosanoids such as prostaglandins and leukotrienes [58]. Odontuya et al. [65] performed the NMR spectroscopic analysis of LUT and its derived glycosides and concluded that their molecular structures could considerably contribute to their anti-inflammatory effects.

Hepatocyte apoptosis occurs through two main pathways: the extrinsic pathway, which depends on signals from the death receptor, and the intrinsic pathway, which is triggered by intracellular stimuli. In agreement with previous studies, Pb exposure was associated with an induction of cell apoptotic mediators and a marked disturbance in the gene expression of apoptosis-regulating proteins [7,47,66]. The apoptosis induced by Pb exposure may result from the damage of the mitochondrial membrane allowing the release of cytochrome-c and caspase-activating factors into the cytosol. Additionally, the potency of Pb in replacing divalent ions, such as calcium, is involved in the imbalance between anti- and pro-apoptotic proteins [47]. LUT pre-treatment significantly reversed the apoptotic events in the hepatocytes by lowering the content of caspase-3 and Bax and increasing Bcl2; this is consistent with findings of other authors [22,46]. Additionally, LUT can rescue the cell from apoptotic changes via increasing expressions of Murine double minute 2 (Mdm2) together with suppression of p53 expression [67]. P53 can induce the transcription-independent apoptosis via triggering the nuclear apoptotic mRNA expression in addition to binding with Bcl-XL and Bcl-2, antiapoptotic proteins with subsequent release of Bak/Bax, pro-apoptotic effectors, from the complex [68]. Mdm2 is one of the most significant regulators of p53 activity by its ubiquitination activity, thereby decreases the stability of p53 [69]. Although the chosen LUT dose was safe and provided protective impacts against PbAc-induced liver damage, previous studies showed that LUT treatment in vitro study at high doses is associated with cytotoxic effects on the primary rat hepatocytes [70]. However, IC50 of LUT against normal cells was increased several-folds compared to tumorigenic cells in study of Kim et al. [71]. Consistent with the used dose, previous studies have used the same dose of LUT without recording oblivious cytotoxic effect [72]. Moreover, Xiao et al. [73] reported that LUT at 100 mg kg^−1^ protected the diabetic rat’s heart against ischemia/reperfusion (I/R) injury without any cytotoxicity. However, further studies are required to evaluate the potential negative outcomes produced following the application of higher doses of LUT on different tissues.

## 5. Conclusions

Collectively, this study illustrated that luteolin could diminish Pb overload in hepatic tissues induced by 1 week of lead acetate injections in male Wistar albino rats. The proposed mechanisms may involve boosting of the anti-oxidant mechanisms, a decrease in the augmented release of inflammatory mediators, and suppression of the apoptotic cascade (Figure 10). Thus, luteolin could be administered as a dietary supplement that protects humans against environmental and occupational lead exposure, without any detrimental side effects.

## Figures and Tables

**Figure 1 antioxidants-09-00010-f001:**
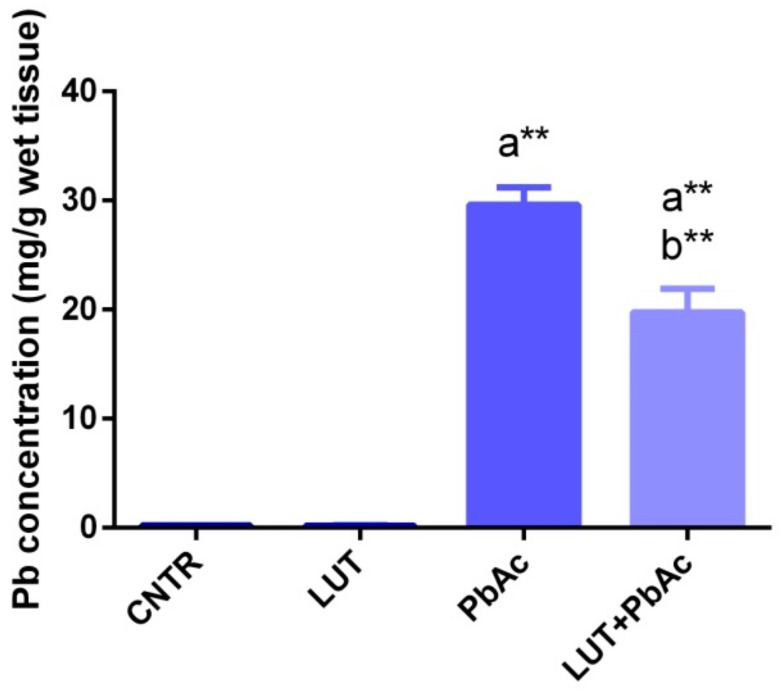
Lead (Pb) residual levels in hepatic tissue following lead acetate (PbAc, 20 mg/kg, i.p.) and/or luteolin (LUT, 50 mg/kg, orally) exposure in male rats. The values represent the means ± SD (*n* = 7). ^a^ represents the statistical significance relative to that of the control group at *p* < 0.05. ^b^ represents the statistical significance relative to that of the PbAc-injected group at *p* < 0.05. ** reflects the statistical significant differences between groups at *p*-values < 0.001.

**Figure 2 antioxidants-09-00010-f002:**
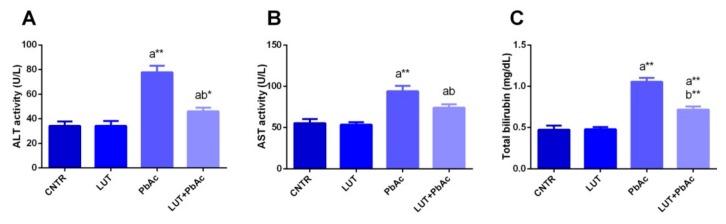
Aspartate aminotransferase (AST) (**A**), alanine aminotransferase (ALT) (**B**), and total bilirubin (**C**) levels following lead acetate (PbAc, 20 mg/kg, i.p.) and/or luteolin (LUT, 50 mg/kg, orally) exposure in male rats. The values are the means ± SD (*n* = 7). ^a^ represents the statistical significance relative to that of the control group at *p* < 0.05. ^b^ represents the statistical significance relative to that of the PbAc-injected group at *p* < 0.05. * reflects the statistical significant differences between groups at *p*-values < 0.01. Moreover, ** reflects the statistical significant differences between groups at *p*-values < 0.001.

**Figure 3 antioxidants-09-00010-f003:**
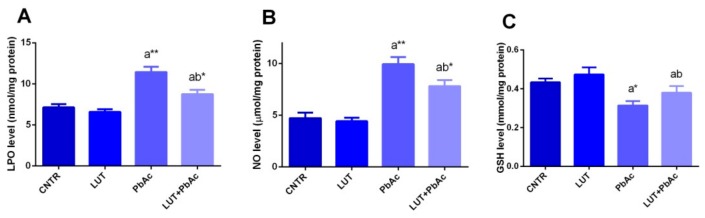
Hepatic levels of lipid peroxidation (LPO; **A**), nitric oxide (NO; **B**), and glutathione (GSH; **C**) following lead acetate (PbAc, 20 mg/kg, i.p.) and/or luteolin (LUT, 50 mg/kg, orally) exposure in male rats. The values are the means ± SD (*n* = 7). ^a^ represents the statistical significance relative to that of the control group at *p* < 0.05. ^b^ represents the statistical significance relative to that of the PbAc-injected group at *p* < 0.05. * reflects the statistical significant differences between groups at *p*-values < 0.01. Moreover, ** reflects the statistical significant differences between groups at *p*-values < 0.001.

**Figure 4 antioxidants-09-00010-f004:**
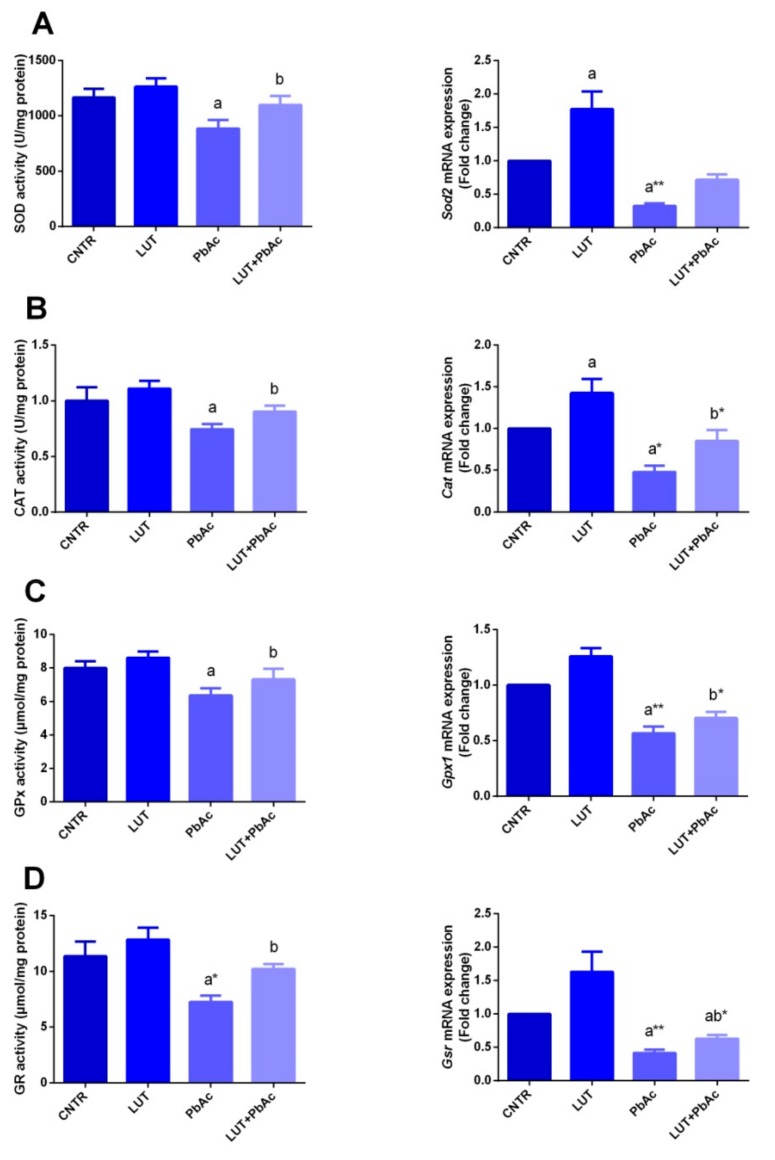
Effects of luteolin (LUT, 50 mg/kg, orally) on the activity of superoxide dismutase (SOD; **A**), and catalase (CAT; **B**), glutathione peroxidase (GPx; **C**), and glutathione reductase (GR; **D**), and their mRNA expression in the liver tissue following lead acetate (PbAc, 20 mg/kg, i.p.). The biochemical data are expressed as the mean ± SD (*n* = 7). mRNA expression results are recorded as the mean ± SD of five assays in duplicate referenced to *Gapdh* and represented as fold changes (log2 scale) as compared with the mRNA levels of the control group. ^a^ represents the statistical significance relative to that of the control group at *p* < 0.05. ^b^ represents the statistical significance relative to that of the PbAc-injected group at *p* < 0.05. * reflects the statistical significant differences between groups at *p*-values < 0.01. Moreover, ** reflects the statistical significant differences between groups at *p*-values < 0.001.

**Figure 5 antioxidants-09-00010-f005:**
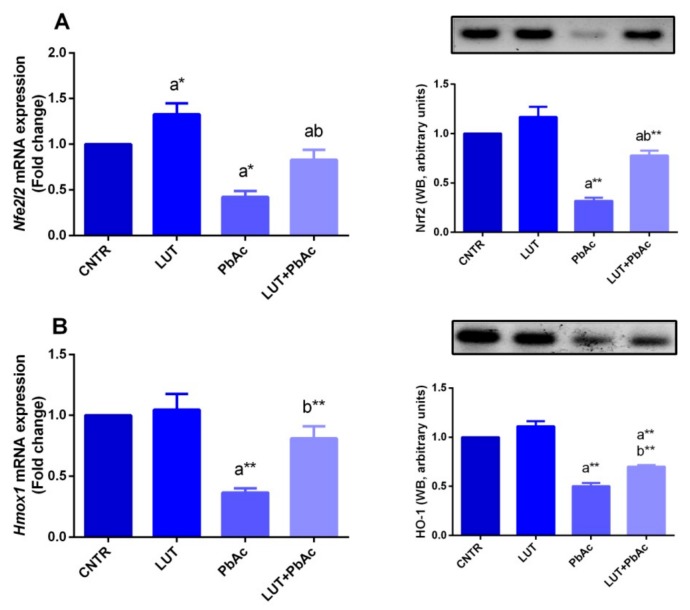
Hepatic protein and mRNA gene expression levels of Nrf2 (**A**) and HO-1 (**B**) following lead acetate (PbAc, 20 mg/kg, i.p.) and/or luteolin (LUT, 50 mg/kg, orally) exposure in male rats. Results of mRNA are recorded as the mean ± SD of five assays in duplicate and *Gapdh* was the housekeeping gene, whereas β-actin was used as the reference control for western blot analysis. ^a^ represents the statistical significance relative to that of the control group at *p* < 0.05. ^b^ represents the statistical significance relative to that of the PbAc-injected group at *p* < 0.05. * reflects the statistical significant differences between groups at *p*-values < 0.01. Moreover, ** reflects the statistical significant differences between groups at *p*-values < 0.001.

**Figure 6 antioxidants-09-00010-f006:**
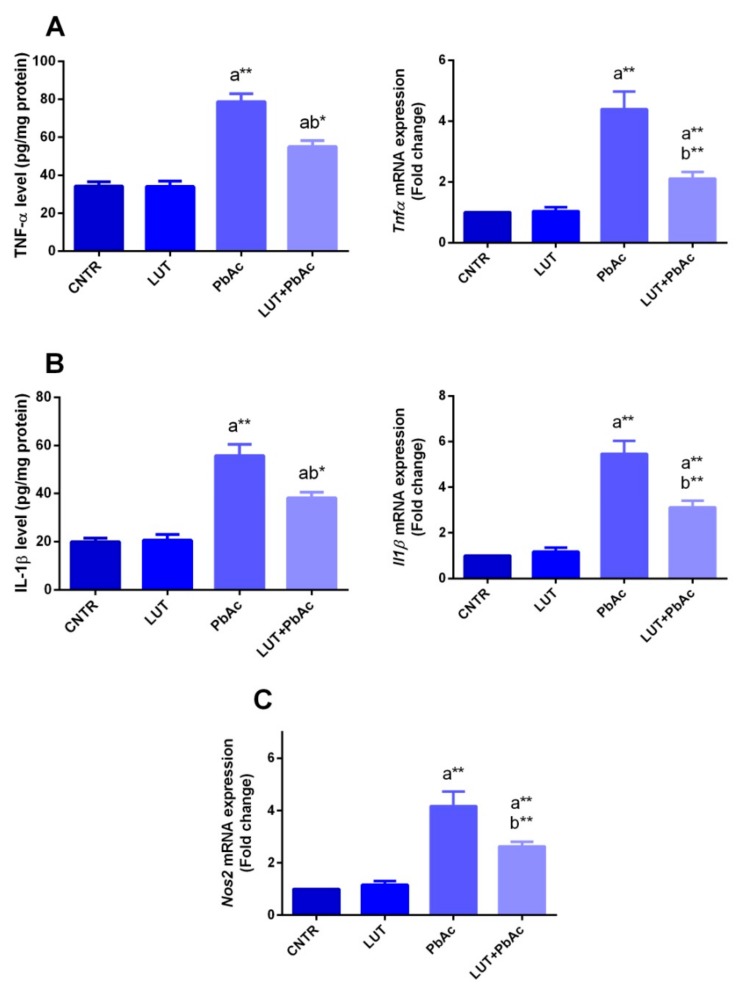
Hepatic levels of TNF-α (**A**) and IL-1β (**B**) and *Nos2* (**C**) mRNA expression following lead acetate (PbAc, 20 mg/kg, i.p.) and/or luteolin (LUT, 50 mg/kg, orally) exposure in male rats. The biochemical data are expressed as the mean ± SD (*n* = 7). mRNA expression results are recorded as the mean ± SD of five assays in duplicate referenced to *Gapdh* and represented as fold changes (log2 scale) as compared with the mRNA levels of the control group. ^a^ represents the statistical significance relative to that of the control group at *p* < 0.05. ^b^ represents the statistical significance relative to that of the PbAc-injected group at *p* < 0.05. * reflects the statistical significant differences between groups at *p*-values < 0.01. Moreover, ** reflects the statistical significant differences between groups at *p*-values < 0.001.

**Figure 7 antioxidants-09-00010-f007:**
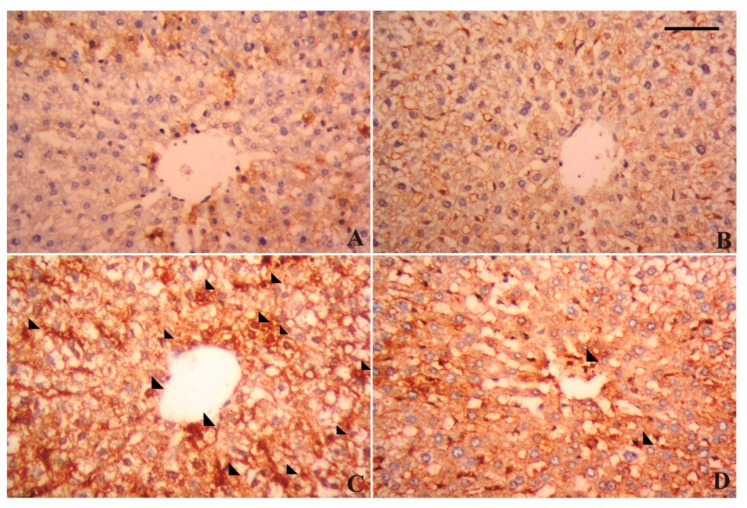
Photomicrographs showing alterations in the hepatic nuclear factor kappa B (NF-κB) expression following lead acetate (PbAc, 20 mg/kg, i.p.) and/or luteolin (LUT, 50 mg/kg, orally) exposure in male rats. (**A**) Control group; (**B**) LUT administered group; (**C**) PbAc treated group; (**D**) LUT+PbAc-treated group; scale bar 80 um.

**Figure 8 antioxidants-09-00010-f008:**
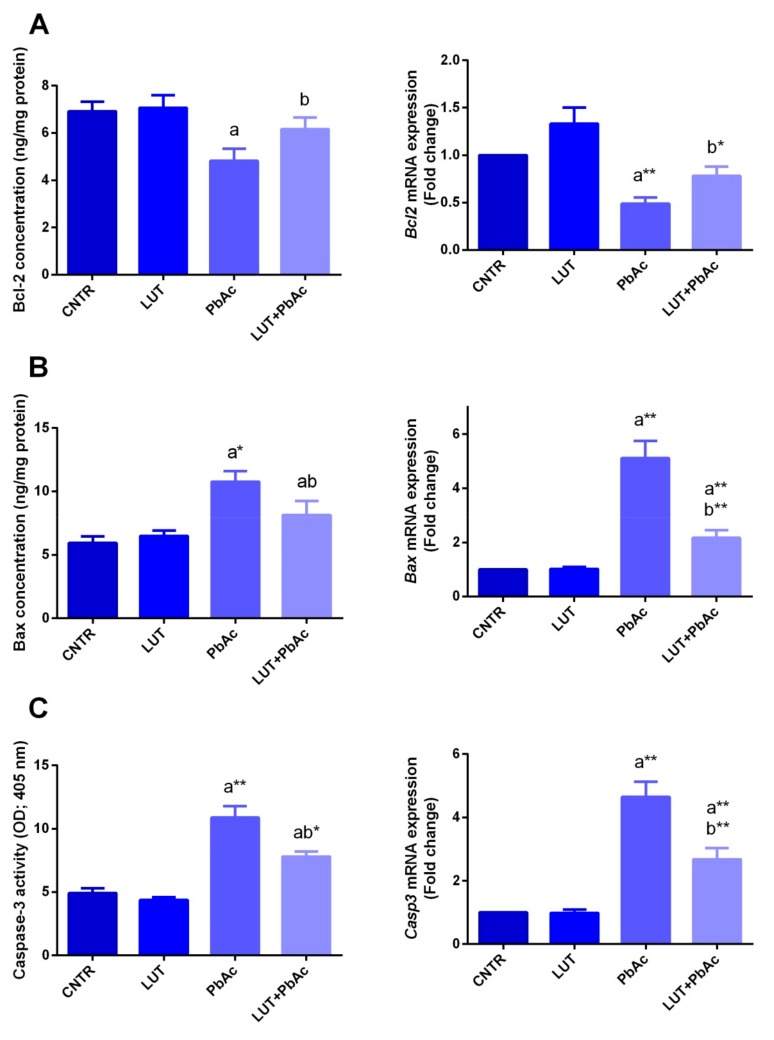
Apoptotic-related proteins (Bcl-2; **A**, Bax; **B** and caspases-3; **C**) level and expression following lead acetate (PbAc, 20 mg/kg, i.p.) and/or luteolin (LUT, 50 mg/kg, orally) exposure in male rats. Bcl-2, Bax and caspases-3 protein levels are expressed as the mean ± SD (*n* = 7). mRNA expression results are recorded as the mean ± SD of five assays in duplicate referenced to *Gapdh* and represented as fold changes (log2 scale) as compared with the mRNA levels of the control group. ^a^ represents the statistical significance relative to that of the control group at *p* < 0.05. ^b^ represents the statistical significance relative to that of the PbAc-injected group at *p* < 0.05. * reflects the statistical significant differences between groups at *p*-values < 0.01. Moreover, ** reflects the statistical significant differences between groups at *p*-values < 0.001.

**Figure 9 antioxidants-09-00010-f009:**
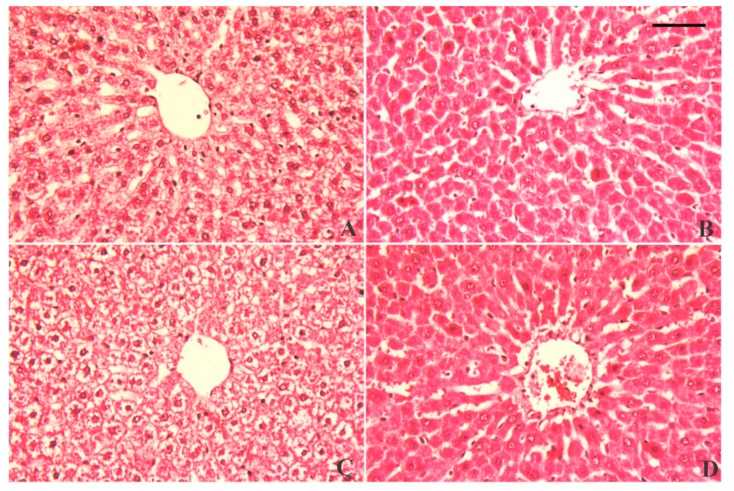
Histopathological alterations of liver tissue following lead acetate (PbAc, 20 mg/kg, i.p.) and/or luteolin (LUT, 50 mg/kg, orally) exposure in male rats. Normal hepatic histological architecture was observed in the control and LUT groups, characterized by normal central veins surrounded by normal, intact hepatocytes (**A** and **B**, respectively). In contrast, PbAc-intoxicated rats exhibited necrotic liver cells associated with considerably degenerated and vacuolated peripheral hepatocytes, along with neutrophil and lymphocyte infiltrations around the peri-portal areas (**C**). However, pretreatment with LUT reversed the histological changes in response to PbAc intoxication (**D**). Scale bar 80 um.

**Figure 10 antioxidants-09-00010-f010:**
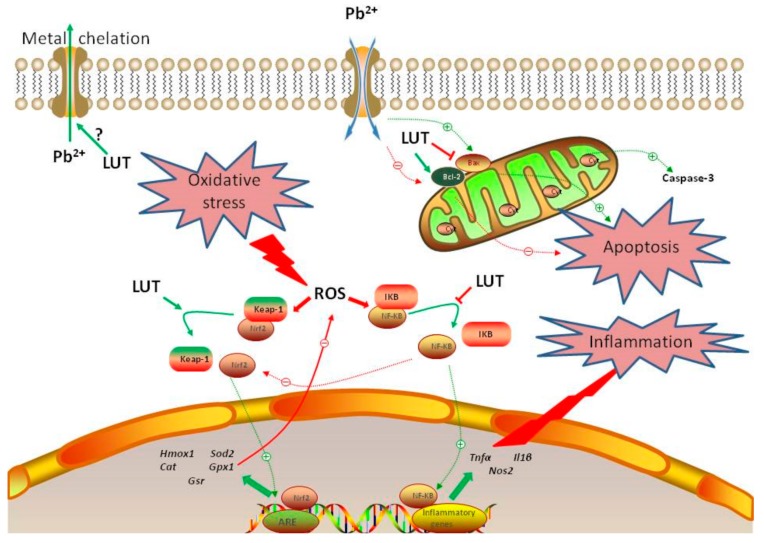
Summary suggesting the mechanisms of LUT attenuated hepatotoxicity induced by PbAc. Pb^2+^ enhanced ROS formation in cytoplasm inducing oxidative stress, finally led to apoptosis and inflammation. However, LUT promoted the Nrf2/ARE pathway and triggered the activation of NF-κB/cell death signaling pathways and prevented liver injury induced by Pb^2+^. Green line denotes stimulatory and red line denotes inhibitory effect.

**Table 1 antioxidants-09-00010-t001:** Primer sequences of genes analyzed in real time polymerase chain reaction (PCR).

Name	Accession Number	Forward Primer (5’---3’)	Reverse Primer (5’---3’)
*Gapdh*	NM_017008.4	AGTGCCAGCCTCGTCTCATA	GATGGTGATGGGTTTCCCGT
*Sod2*	NM_017051.2	TAAGGGTGGTGGAGAACCCA	TGATGACAGTGACAGCGTCC
*Cat*	NM_012520.2	TTTTCACCGACGAGATGGCA	AAGGTGTGTGAGCCATAGCC
*Gpx1*	NM_030826.4	CAGTCCACCGTGTATGCCTT	GTAAAGAGCGGGTGAGCCTT
*Gsr*	NM_053906.2	TACTGCACTTCCCGGTAGGA	TGGATGCCAACCACCTTCTC
*Nfe2l2*	NM_031789.2	TTGTAGATGACCATGAGTCGC	ACTTCCAGGGGCACTGTCTA
*Hmox1*	NM_012580	GCGAAACAAGCAGAACCCA	GCTCAGGATGAGTACCTCCCA
*Nos2*	NM_012611.3	GTTCCTCAGGCTTGGGTCTT	TGGGGGAACACAGTAATGGC
*Tnf*	NM_012675.3	GGCTTTCGGAACTCACTGGA	CCCGTAGGGCGATTACAGTC
*Il1β*	NM_031512.2	GACTTCACCATGGAACCCGT	GGAGACTGCCCATTCTCGAC
*Bcl2*	NM_016993	ACTCTTCAGGGATGGGGTGA	TGACATCTCCCTGTTGACGC
*Bax*	NM_017059.2	GGGCCTTTTTGCTACAGGGT	TTCTTGGTGGATGCGTCCTG
*Casp3*	NM_012922.2	GAGCTTGGAACGCGAAGAAA	TAACCGGGTGCGGTAGAGTA

The abbreviations of the genes; *Gapdh*, glyceraldehyde-3-phosphate dehydrogenase; *Sod2*, superoxide dismutase 2 mitochondrial (MnSOD); *Cat*, catalase; *Gpx1*, glutathione peroxidase 1; *Gsr*, glutathione reductase; *Nfe2l2*, nuclear factor erythroid 2-related factor 2; *Hmox1*: heme oxygenase 1; *Nos2*, inducible nitric oxide synthase; *Tnf*, tumor necrosis factor; *Il1β*, interleukin 1 beta; *Bcl2*: B-cell lymphoma 2; *Bax*, Bcl-2-like protein 4; *Casp3*, caspase-3.

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
