# Peer review of "Antagonistic Efficacy of Luteolin against Lead Acetate Exposure-Associated with Hepatotoxicity is Mediated via Antioxidant, Anti-Inflammatory, and Anti-Apoptotic Activities"

_antioxidants, 2019, doi:10.3390/antiox9010010_

Round 1
Reviewer 1 Report
The present manuscript describes a study reporting health benefits of the flavonoid luteolin in the mitigation of liver damage induced by the exposure to the heavy metal lead (lead acetate).
The study is interesting, well-performed, incorporating many data. The data are well-presented and the manuscript is well-written.
However, some issues should be addressed and could be improved
1. Last sentence in the Abstract: It is more accurate to mention that your findings describe potential mechanisms involved in the alleviation......
Pretreatment of Luteolin for 3 hours resulted in the alleviation of liver injury, luteolin may have protected cells from the damage induced by Pb. It would have been nice (in a following study) to have data on the effects of luteolin after liver injury induced by Pb.
2. I would suggest to incorporate the study limitations text in the discussion and discuss them. It is important to include how many times lower dose of Luteolin was used in your experiments as compared to the high dose that induces cytotoxic effects (ref. 63 and 64) on primary hepatocytes (exrapolation).
3. In the figures 2, 3, 4, 5, 6, 8 please add A, B, C, D...for each diagram and in the legend the description. Please provide a short title in each figure legend.
4. line 129: please remove using ELISA (written two times)
5. line 227: ... remarkably reversed when....: please see point 1, it is not clear whether luteolin reverses the effects or protects cells from the damaging effects.
Author Response
Reviewer 1:
The present manuscript describes a study reporting health benefits of the flavonoid luteolin in the mitigation of liver damage induced by the exposure to the heavy metal lead (lead acetate). The study is interesting, well-performed, incorporating many data. The data are well-presented and the manuscript is well-written.
However, some issues should be addressed and could be improved
Last sentence in the Abstract: It is more accurate to mention that your findings describe potential mechanisms involved in the alleviation......Response: Thank you for your comment; we have followed your comment.
Pretreatment of Luteolin for 3 hours resulted in the alleviation of liver injury, luteolin may have protected cells from the damage induced by Pb. It would have been nice (in a following study) to have data on the effects of luteolin after liver injury induced by Pb.
Response: Thank you for your comment; we agree with you that evaluating the effect of post treatment with luteolin will give more details about how luteolin can protect liver tissue upon lead toxicity and maybe we can perform this experiment in the soon future.
I would suggest to incorporate the study limitations text in the discussion and discuss them. It is important to include how many times lower dose of Luteolin was used in your experiments as compared to the high dose that induces cytotoxic effects (ref. 63 and 64) on primary hepatocytes (exrapolation).Response: Thank you for your comment; we have followed your comment.
In the figures 2, 3, 4, 5, 6, 8 please add A, B, C, D...for each diagram and in the legend the description. Please provide a short title in each figure legend.Response: Thank you for your comment; we have followed your comment.
line 129: please remove using ELISA (written two times)Response: Thank you for your comment; we have followed your comment.
line 227: ... remarkably reversed when....: please see point 1, it is not clear whether luteolin reverses the effects or protects cells from the damaging effects.Response: Thank you for your comment; we think that luteolin protected the liver tissue against PbAc intoxication as it given prior to PbAc, therefore we replaced the word reversed with protected.
Reviewer 2 Report
This study has looked at hepatotoxicity induced in rats by exposure to high levels of lead and evaluated the potential of the flavonoid luteolin to ameliorate adverse changes. Lead-linked liver dysfunction was found to be associated with increased apoptosis and inflammation and disruption to antioxidant function. Oral administration of luteolin limited this liver disruption.
The authors propose that this luteolin protective effect may be due boosting of the antioxidant functions, a decreased release of inflammatory mediators, and suppression of the apoptotic cascade. While this may be the case, there could be a simpler explanation in that luteolin binds free lead and facilitates its rapid clearance via bile and ultimately feces or in the urine. If less free lead reached or persisted in the liver, the disruption and damage in the organ would be less, which is in line with the data presented. Unfortunately, daily fecal and urine excretion of lead has not been monitored. The authors need to provide this information or at least explain why they do not think it is a factor.
While the discussion comprehensively fits the biological and physiological results of the present study on lead poisoning with those already in the literature, the impression for the reader is that most of the work is confirmatory and that the only novel finding is that luteolin ameliorates the effects of lead. There needs to be a more extensive discussion of the possible mechanisms of action of luteolin and its potential for prophylactic or therapeutic use.
There is a need to clarify how the statistical differences are indicated in figures and tables. At present in all, there is an indication that ‘a - represents the statistical significance relative to that of the control group at p<0.05. b represents the statistical significance relative to that of the PbAc-injected group at p<0.05’ but some Pb+LUT columns have a & b above each other while some just have ‘ab’ shown. Please clarify. Also, what is the meaning of the asterisks on the figures? They are not mentioned in the legends or the general text.
Ln 93-99 There is a need to clarify that all animals were injected and orally dosed with test or control substances daily. As it reads at present group II were only dosed orally and group III were injected only.
The length of the treatment must be stated in the method.
Ln 101-109 Why were no other tissues, such as kidney, collected and evaluated
Ln 105-107 How was the tissue dispersed?
Ln 120-122 Source of kits used in the assays.
Ln 198-199 Why are no data given? Should there not be a link to supplemental table 1? If food and water intake monitored why not fecal and urine excretion?
Ln 217-220 See general comments on statistical differences
Ln 234-237
Ln 239-246
Ln 257-262
Ln 275-281
Ln 296-302
Ln 251 ‘upregulation’ Are the factors upregulated or is it that the loss of these factors is diminished by LUT?
Ln 323-330 Repeats information in the introduction.
Author Response
Reviewer 2:
This study has looked at hepatotoxicity induced in rats by exposure to high levels of lead and evaluated the potential of the flavonoid luteolin to ameliorate adverse changes. Lead-linked liver dysfunction was found to be associated with increased apoptosis and inflammation and disruption to antioxidant function. Oral administration of luteolin limited this liver disruption.
The authors propose that this luteolin protective effect may be due boosting of the antioxidant functions, a decreased release of inflammatory mediators, and suppression of the apoptotic cascade. While this may be the case, there could be a simpler explanation in that luteolin binds free lead and facilitates its rapid clearance via bile and ultimately feces or in the urine. If less free lead reached or persisted in the liver, the disruption and damage in the organ would be less, which is in line with the data presented. Unfortunately, daily fecal and urine excretion of lead has not been monitored. The authors need to provide this information or at least explain why they do not think it is a factor.
While the discussion comprehensively fits the biological and physiological results of the present study on lead poisoning with those already in the literature, the impression for the reader is that most of the work is confirmatory and that the only novel finding is that luteolin ameliorates the effects of lead. There needs to be a more extensive discussion of the possible mechanisms of action of luteolin and its potential for prophylactic or therapeutic use.
Response: Thank you for your comment; we have added a number of subsections that comprehensively illustrate the various protective mechanisms of luteolin in the discussion including its chelating ability, antioxidant, anti-inflammatory and anti-apoptotic activities.
There is a need to clarify how the statistical differences are indicated in figures and tables. At present in all, there is an indication that ‘a - represents the statistical significance relative to that of the control group at p<0.05. b represents the statistical significance relative to that of the PbAc-injected group at p<0.05’ but some Pb+LUT columns have a & b above each other while some just have ‘ab’ shown. Please clarify. Also, what is the meaning of the asterisks on the figures? They are not mentioned in the legends or the general text.
Response: Thank you for your comment; we have added the meaning of ab and defined the meanings of the used asterisks on the figures in the legends.
Ln 93-99: There is a need to clarify that all animals were injected and orally dosed with test or control substances daily. As it reads at present group II were only dosed orally and group III were injected only.
Response: Thank you for your comment; we have modified this section to meet your suggestion.
The length of the treatment must be stated in the method.
Response: Thank you for your comment; we have followed your comment.
Ln 101-109: Why were no other tissues, such as kidney, collected and evaluated.
Response: Thank you for your comment; we have actually collected other organs including brain, kidney and testis for further studies.
Ln 105-107: How was the tissue dispersed?
Response: Thank you for your comment; tissue collection and dissection were mentioned between lines 105-112.
Ln 120-122: Source of kits used in the assays.
Response: Thank you for your comment; we have followed your comment.
Ln 198-199: Why are no data given? Should there not be a link to supplemental table 1? If food and water intake monitored why not fecal and urine excretion?
Response: Thank you for your comment; we have added the food and water intake/group as a supplementary table in the revised version. Concerning the fecal and urine excretion we don’t have in our lab the metabolic cage to monitor these factors.
Ln 217-220: See general comments on statistical differences
Ln 234-237
Ln 239-246
Ln 257-262
Ln 275-281
Ln 296-302
Response: Thank you for your comment; we have modified the figures legends to meet your suggestion.
Ln 251: ‘upregulation’ Are the factors upregulated or is it that the loss of these factors is diminished by LUT?
Response: Thank you for your comment; the expression of Nrf2 and HO-1 were downregulated following PbAc exposure, while LUT upregulated these antioxidant promoters in response to PbAc.
Ln 323-330: Repeats information in the introduction.
Response: Thank you for your comment; we have deleted this paragraph to avoid the repetition.
Round 2
Reviewer 2 Report
The authors have dealt with all matters raised and queries in a satisfactory manner.
This manuscript is a resubmission of an earlier submission. The following is a list of the peer review reports and author responses from that submission.
Round 1
Reviewer 1 Report
The manuscript (Ms) by Alkhuriji et al. describes hepatotoxic effects of lead (Pb) and the detoxifying effects of a natural flavone luteolin in Wistar rats. The finding that different natural flavonoids (or natural mixtures) confer protection against heavy metals intoxication in liver and other organs has been already documented by several earlier papers. The novelty of the present Ms is limited to the combination of luteolin and Pb-induced liver damage, while both the hepatotoxicity of Pb and hepatoprotection of luteolin against heavy metals have been already shown in previous papers (cited in Ms). The data presented in this Ms confirm that Pb accumulation in the liver causes an upregulation of lipid peroxidation, liver toxicity markers AST and ALT and inflammatory markers, while luteolin attenuates these parameters. The authors focus on the characterization of mRNA levels and activities of several antioxidant enzymes in the liver and show, in accordance with previous reports, a surprising downregulation of transcripts coding for SOD, Glutathione Reductase and GSH Peroxidase, NRF2 and other antioxidant proteins. These data are corroborated by the expected downregulation of their enzymatic activities.
The MS is well-organized and clear, but contains some critical incongruences in the methods, data and numbers of animals that are difficult to reconcile. There are also gaps in the discussion and additional data would be needed to increase the novelty of this work. The major concerns are pointed out below:
Major points:
The authors used 50 mg/kg of luteolin, which is around 500 ul of 20 mg/ml solution per animal. This is a quite a high dose and concentration of this lipophilic flavone. The problem is that they do not explain how it was dissolved. Luteolin is insoluble in saline solution which was administered to the control group. The authors cite Kabolandi et al. who used 1% DMSO to partially solubilize luteolin. However, DMSO was missing in the control group saline solution and was not mentioned at all in the materials and methods section. Table 1 reporting primer sequences is missing. The authors claim they used 7 animals per group, but the declared total number of rats was 24. The authors should attach the raw data from real-time RT-PCR analysis for two representative antioxidant enzymes mRNAs and for Tnf-alpha mRNA (for 7 animals each exp. group) as externally available data set or supplementary material. The normal AST values in adult Wistar rats are much higher than in humans and are around 100 – 120 U/L (Yang D et al. Sci Rep 2016, Arhoghro et al. Eur J Sci Res 2009). How would the authors explain lower AST values (50 U/L) in control and luteolin treated animals (Fig. 2)? Pure flavones are poorly absorbed by the intestine. Can the authors prove that luteolin gets to the blood stream? What is the amount of absorbed luteolin with respect to total luteolin in control (luteolin alone) and Pb-treated rats? At least the quantification of luteolin and its metabolites in the blood and in the liver tissue should be added to the final version of the Ms. Western blotting data with total protein levels for at least 3 representative rats from each experimental group should be presented for Nfkb, Bcl-2, Bax and possibly Caspase-2 to corroborate data in Fig. 7 and 8. The protein levels of Bcl-2, Bax and Caspase-2 activity (Fig. 8) have been apparently quantified by an unknown method. These methods are not mentioned and described anywhere in the Ms. These methods must be described in the materials and methods section and mentioned in the figure legend. Luteolin is known to be highly toxic in both cancer and primary, non-transformed cells in cell culture, including primary rat hepatocytes (Hytii et al. J Nutr Biochem 2017, Lascala et al. J Nutr Biochem 2018, Shi F et al. J Appl Toxicol. 2015) How the authors would explain the hepatoprotective effect of luteolin in vivo? This discrepancy should be discussed.Minor points:
The method to sacrifice the rats has not been described in the materials and methods section and this should be corrected. Did the animal body mass change as the result of Pb intoxication, luteolin treatment? The body mass data before and after the experiment should be shown. The authors should suggest what would be the mechanism of Pb-induced transcriptional downregulation of antioxidant enzymes as this is an unexpected finding. The authors should attach the raw data from real-time RT-PCR analysis for two representative antioxidant enzymes mRNAs and for Tnf-alpha mRNA (for 7 animals each exp. group) as externally available data set or supplementary material.Author Response
Reviewer 1:
Comments and Suggestions for Authors
The manuscript (Ms) by Alkhuriji et al. describes hepatotoxic effects of lead (Pb) and the detoxifying effects of a natural flavone luteolin in Wistar rats. The finding that different natural flavonoids (or natural mixtures) confer protection against heavy metals intoxication in liver and other organs has been already documented by several earlier papers. The novelty of the present Ms is limited to the combination of luteolin and Pb-induced liver damage, while both the hepatotoxicity of Pb and hepatoprotection of luteolin against heavy metals have been already shown in previous papers (cited in Ms). The data presented in this Ms confirm that Pb accumulation in the liver causes an upregulation of lipid peroxidation, liver toxicity markers AST and ALT and inflammatory markers, while luteolin attenuates these parameters. The authors focus on the characterization of mRNA levels and activities of several antioxidant enzymes in the liver and show, in accordance with previous reports, a surprising downregulation of transcripts coding for SOD, Glutathione Reductase and GSH Peroxidase, NRF2 and other antioxidant proteins. These data are corroborated by the expected downregulation of their enzymatic activities.
The MS is well-organized and clear, but contains some critical incongruences in the methods, data and numbers of animals that are difficult to reconcile. There are also gaps in the discussion and additional data would be needed to increase the novelty of this work. The major concerns are pointed out below:
Major points:
The authors used 50 mg/kg of luteolin, which is around 500 ul of 20 mg/ml solution per animal. This is a quite a high dose and concentration of this lipophilic flavone. The problem is that they do not explain how it was dissolved. Luteolin is insoluble in saline solution which was administered to the control group. The authors cite Kabolandi et al. who used 1% DMSO to partially solubilize luteolin. However, DMSO was missing in the control group saline solution and was not mentioned at all in the materials and methods section.
Response: Thanks for your comment; the chosen dose of LUT used in the present study was safe as it used in a previous study at higher dose rich to 100 mg/kg (PMID: 27444056). In addition, we have explained the missed details about how LUT was dissolved in the revised version in chemicals and reagents sections as follow:
LUT was first dissolved in dimethyl sulfoxide (DMSO) and then diluted with normal saline (0.9% NaCl); the final administered solution contained 5% DMSO, corresponding to 20 μl per rat. Control rats received the same dose of DMSO (5% DMSO in saline). Although DMSO could represent free radical scavenger effect, its use was necessary because LUT is hydrophobic. It was therefore used at the lowest level possible.
Table 1 reporting primer sequences is missing.
Response: Thanks for your comment; we have added the missing table.
The authors claim they used 7 animals per group, but the declared total number of rats was 24. The authors should attach the raw data from real-time RT-PCR analysis for two representative antioxidant enzymes mRNAs and for Tnf-alpha mRNA (for 7 animals each exp. group) as externally available data set or supplementary material.
Response: Thanks for your comment; we have corrected the mistakes in total used number of animals and the figure legends, as we used 7 measurements for the biochemical estimations and only just 3 runs for RT-PCR analysis (not 7). However, the obtained raw data is added as a supplementary material.
The normal AST values in adult Wistar rats are much higher than in humans and are around 100 – 120 U/L (Yang D et al. Sci Rep 2016, Arhoghro et al. Eur J Sci Res 2009). How would the authors explain lower AST values (50 U/L) in control and luteolin treated animals (Fig. 2)?
Response: Thanks for your comment; the obtained AST level is in consistence with/or close to several previous studies (PMID: 27795452; PMID: 20587623).
Pure flavones are poorly absorbed by the intestine. Can the authors prove that luteolin gets to the blood stream? What is the amount of absorbed luteolin with respect to total luteolin in control (luteolin alone) and Pb-treated rats? At least the quantification of luteolin and its metabolites in the blood and in the liver tissue should be added to the final version of the Ms.
Response: Thanks for your comment; unfortunately we don’t have fund to repeat the experiment plus we don’t have the facility to estimate the level of LUT in the circulation as well.
Western blotting data with total protein levels for at least 3 representative rats from each experimental group should be presented for Nfkb, Bcl-2, Bax and possibly Caspase-2 to corroborate data in Fig. 7 and 8.
Response: Thanks for your comment; we have evaluated the protein levels using ELISA method and the method has been added to the revised version. Currently we don’t have the facility to estimate these proteins using Western blotting which will for sure be more valuable.
The protein levels of Bcl-2, Bax and Caspase-2 activity (Fig. 8) have been apparently quantified by an unknown method. These methods are not mentioned and described anywhere in the Ms. These methods must be described in the materials and methods section and mentioned in the figure legend.
Response: Thanks for your comment; we have evaluated the protein levels using ELISA method and the method has been added to the revised version.
Luteolin is known to be highly toxic in both cancer and primary, non-transformed cells in cell culture, including primary rat hepatocytes (Hytii et al. J Nutr Biochem 2017, Lascala et al. J Nutr Biochem 2018, Shi F et al. J Appl Toxicol. 2015) How the authors would explain the hepatoprotective effect of luteolin in vivo? This discrepancy should be discussed.
Response: Thanks for your comment; previous studies demonstrated that cancer cells of human colon cancer (HT-29 and SNU-407), HeLa and U2OS are more sensitive to LUT and the IC50 were increased several-folds compared to tumorigenic cells.
Minor points:
The method to sacrifice the rats has not been described in the materials and methods section and this should be corrected.
Response: Thanks for your comment; the method has been added in the revised version.
Did the animal body mass change as the result of Pb intoxication, luteolin treatment? The body mass data before and after the experiment should be shown. The authors should suggest what would be the mechanism of Pb-induced transcriptional downregulation of antioxidant enzymes as this is an unexpected finding.
Response: Thanks for your comment; the mechanism involved in Pb-induced transcriptional downregulation of antioxidant enzymes has been added to the revised Ms.
The authors should attach the raw data from real-time RT-PCR analysis for two representative antioxidant enzymes mRNAs and for Tnf-alpha mRNA (for 7 animals each exp. group) as externally available data set or supplementary material.
Response: Thanks for your comment; we have corrected the mistakes in the figure legends as we used 7 measurements for the biochemical estimations and only just 3 runs for RT-PCR analysis (not 7). However, the obtained raw data is added as a supplementary material.

Reviewer 2 Report
The authors are reporting, in this manuscript, a detailed mechanism for the alleviation of lead-induced liver injury by luteolin via its potent anti-inflammatory, antioxidant, and anti-apoptotic properties.
The study of the effect of lead exposure in Wistar rats is well investigated by several groups.
e.g., Toxic effects of lead exposure in Wistar rats: involvement of oxidative stress and the beneficial role of edible jute (Corchorus olitorius) leaves.
https://www.ncbi.nlm.nih.gov/pubmed/23291325
Also Luteolin is a well know commercially available antioxidant compound that was reported to reduce an oxidative insult by oxidative stress inducers (isoproterenol) https://www.sciencedirect.com/science/article/pii/S2211558712000313?via%3Dihub
Based on the above statements and while this manuscript is very significant it is lucking, in my opinion, novelty which lower its promise.
Additional major comments:
The lipid repair enzymes Glycoperoxidase are a family of 8 enzymes. I am wondering why the authors selected only to evaluate GPX1.
Line 171: TO clearly understand the effect on Lead on the rats appetite, it is recommended to show the body weight data
Line 194: the study conducted for 7 days. Lead may have a long-term effect in the body. I am wondering what the outcome of this study will be for 30 days?
In this study, the authors used LUT-treated group (50 mg/kg, oral, daily), PbAc-treated group (20 mg/kg, i.p., daily), It seems that LUT is not able to entirely mitigate the oxidative damage induced by lead. As the dietary supplement is not too toxic, I am wondering, what happen if we increase the concentration of LUT to 100-150 mg/kg.
The figures 7 and 9 luck more details and need better zoomed images.
For mechanism insight, the authors may need to be aware of the recent reported 2019 study:
Complexation of luteolin with lead (II): Spectroscopy characterization and theoretical researches. https://www.ncbi.nlm.nih.gov/pubmed/30669063
Minor:
Some abbreviation needs to be defined as soon as it is cited first (for example:
LINE 24: ALT and AST
LINE 115: The enzyme activity of ALT and AST
LINE 183: ALT and AST
Confusion between the number of animals used for the entire experiment
Line 78: Twenty-four male Wister albino rats, 10 weeks old and weighing 180–220 g
Lines 88-89: report the use of 4 groups of 7 animals.
Author Response
Reviewer 2:
Comments and Suggestions for Authors
The authors are reporting, in this manuscript, a detailed mechanism for the alleviation of lead-induced liver injury by luteolin via its potent anti-inflammatory, antioxidant, and anti-apoptotic properties.
The study of the effect of lead exposure in Wistar rats is well investigated by several groups. e.g., Toxic effects of lead exposure in Wistar rats: involvement of oxidative stress and the beneficial role of edible jute (Corchorus olitorius) leaves.
https://www.ncbi.nlm.nih.gov/pubmed/23291325
Also Luteolin is a well know commercially available antioxidant compound that was reported to reduce an oxidative insult by oxidative stress inducers (isoproterenol) https://www.sciencedirect.com/science/article/pii/S2211558712000313?via%3Dihub.
Based on the above statements and while this manuscript is very significant it is lucking, in my opinion, novelty which lower its promise.
Additional major comments:
The lipid repair enzymes Glycoperoxidase are a family of 8 enzymes. I am wondering why the authors selected only to evaluate GPX1.
Response: Thanks for your comment; GPx1 has been chosen because it is the main dominant member in the hepatic tissue more than the other enzymes (PMID: 10781391).
Line 171: TO clearly understand the effect on Lead on the rats appetite, it is recommended to show the body weight data
Response: Thanks for your comment; body weight results were attached as a supplementary data.
Line 194: the study conducted for 7 days. Lead may have a long-term effect in the body. I am wondering what the outcome of this study will be for 30 days?
Response: Thanks for your comment; we just focused on evaluating the acute toxicity of lead and we may investigate the effect of longer course exposure in the future.
In this study, the authors used LUT-treated group (50 mg/kg, oral, daily), PbAc-treated group (20 mg/kg, i.p., daily), It seems that LUT is not able to entirely mitigate the oxidative damage induced by lead. As the dietary supplement is not too toxic, I am wondering, what happen if we increase the concentration of LUT to 100-150 mg/kg.
Response: Thanks for your comment; according to the obtained statistical analysis, LUT was able significantly to minimize oxidative challenge in the hepatic tissue following PbAc intoxication. However, LUT supplementation with higher dose and/or longer duration may produce better results with minimal side effects, but for sure, more studies are required.
The figures 7 and 9 luck more details and need better zoomed images.
Response: Thanks for your comment; we added arrow to demonstrate the translocation of NF-κB from the cytoplasm to the nucleus as you suggested.
For mechanism insight, the authors may need to be aware of the recent reported 2019 study:
Complexation of luteolin with lead (II): Spectroscopy characterization and theoretical researches. https://www.ncbi.nlm.nih.gov/pubmed/30669063
Response: Thanks for your comment; we have used the mentioned report.
Minor:
Some abbreviation needs to be defined as soon as it is cited first (for example:
LINE 24: ALT and AST
LINE 115: The enzyme activity of ALT and AST
LINE 183: ALT and AST
Response: Thanks for your comment; we have followed your suggestion.
Confusion between the number of animals used for the entire experiment
Line 78: Twenty-four male Wister albino rats, 10 weeks old and weighing 180–220 g
Lines 88-89: report the use of 4 groups of 7 animals.
Response: Thanks for your comment; we have corrected the mistake in the total used number to avoid any confusion in the revised version.
Round 2
Reviewer 1 Report
The manuscript has been partially improved, but some of explanations provided but authors are not convincing. None of the requested experiments have been done and only final rats weight was provided. The authors did not understand the major point 9 about luteolin toxicity in primary rat hepatocytes. I guess the authors did not check the cited papers (Hytii et al. J Nutr Biochem 2017, Lascala et al. J Nutr Biochem 2018, Shi F et al. J Appl Toxicol. 2015) and answered out of the context.
I could understand to a certain point that the authors cannot make some of the requested experiments due to lack of funding as they declare.
However, I cannot understand and accept that the authors declare in the paper 7 animals per group, confirm it in the figure legend and the truth is that they performed the RT-PCR analysis only for 3 animals per group. The authors response is: "Response: Thanks for your comment; we have corrected the mistakes in total used number of animals and the figure legends, as we used 7 measurements for the biochemical estimations and only just 3 runs for RT-PCR analysis (not 7). However, the obtained raw data is added as a supplementary material."
First, "n=7" has not been corrected to "n=3 for RT-PCR data" in the relative figure legends. In addition the RT-PCR analysis is not in triplicate for each point (as declared in Material and Methods), but only in duplicate in raw data. My main concern is that, the difference between Pb and Pb+LUT groups cannot be statistically significant with 3 points per group by any ANOVA test, for majority of analyzed genes, based on the provided raw data. Finally, SD values do not correspond to SD values presented in figures.
These issues bring all mistakes and discrepancies over the level of tolerance.
Reviewer 2 Report
The manuscript requires further editing.